# Transition approaches for autistic young adults: A case series study

Yosheen Pillay[1]*, Charlotte Brownlow[2], Sonja March[3]

1 School of Education, Centre for Health Research, University of Southern Queensland, Springfield, Queensland, Australia, 2 Centre for Health Research, University of Southern Queensland, Toowoomba, Queensland, Australia, 3 Centre for Health Research, School of Psychology and Wellbeing, University of Southern Queensland, Springfield, Queensland, Australia

These authors contributed equally to this work.

* Yosheen.Pillay@usq.edu.au

## Abstract

The aim of this study was to evaluate the experience of autistic young adults aged 18 to 25 years old over a 12-month transition period from 2016 to 2017. Data was collected through a longitudinal repeated measures case series design with assessments conducted at 2 time points, at baseline then 12 months later. Assessments included self-report evaluations of transition planning and intervention received at high school, engagement in post-secondary education and access to employment, living circumstances, and social support. Examination of 9 cases showed family and social support was an important facilitator of successful transition whilst low independence was a risk factor associated with unsuccessful transition. In-depth analysis of cases showed a lack of engagement in post-secondary education and unemployment were associated with poor quality of life whilst skills development, work experience placements, and support from service providers were associated with improved quality of life. Implications of the findings highlight the need for educational and socially inclusive interventions to support the heterogeneity in individual, social, communication, and behavioural challenges in autistic young adults.

## Introduction

Autism is estimated to affect 1 in 52 Australian adolescents aged 13 to 15 years old, with an overall increase of 42% in the number of Australians diagnosed since 2012 [1,2]. It is reported that autism is more prevalent in males than females estimated at 4 males to every one female [3], but studies have questioned whether the ratio may be closer than this [4]. Autistic individuals experience variability in social functioning and communication with each person demonstrating different strengths and challenges [5]. The majority experience challenges in education, employment, independent living, friendships, and romantic relationships throughout their adult lives [6,7]. The growing prevalence of autistic young adults would suggest that a significant group within the community faces a lifetime of disadvantage because of lack of support for their autism. Difficulty coping with change and adjusting to new environments is widely considered a hallmark of autism. As such, independent living and access to

**Data Availability Statement:** The data underlying the results presented in the study are available from https://dataverse.harvard.edu/dataset.xhtml?persistentId=doi:10.7910/DVN/SOY24P&version=DRAFT.

**Funding:** The authors received no specific funding for this work.

**Competing interests:** The authors have declared that no competing interests exist.

employment associated with the transition to adulthood may be compounded with multiple difficulties unique to autism. Autistic females may experience increased risk as they have lower employment rates and earnings, work fewer hours, and have an increased dependence on disability support when compared to males [8].

Transition indicates a shift in role status for young adults aged 18 to 25 years from an adolescent to undertaking adult responsibilities within society [9]. Markers of a successful transition to adulthood include high school completion, post-secondary education, employment, independence, integrating with the community, and good personal and social relationships [10–12]. Successfully transitioning from high school to post-secondary education can improve future employment options, increase financial independence, and improve quality of life (QoL) in adulthood [13–15]. Several studies have documented poor post-secondary outcomes for autistic young adults [13,16]. Compared to adults with other disabilities, post-school outcomes are the poorest for autistic young adults who have the lowest employment rates and highest rate of no activities after school [11,17]. These findings suggest that targeted transition planning for autistic young adults is needed to address barriers to participation in post-secondary education and employment. Transition planning is recognised as a collaborative effort across interagency providers, schools, individuals, and families [18]. However, recent findings indicate insufficient interagency involvement, limited skill building and community access restricted efforts to gain employment [19].

An individual is more likely to experience a high QoL if important needs in major life settings of education, work, home, and community are fulfilled [20,21]. Involvement in social networks and having people to talk to, is closely linked to physical and mental wellbeing and can contribute to improved QoL during the transition to adulthood [22]. However, social interaction difficulties as experienced by autistic young adults may lead to withdrawal and social isolation [23]. Such individuals may be prone to social anxiety due to struggles in communication and limited ability to socialise with others [24].

QoL is defined as an individual's position in relation to their culture, value system, goals, standards, expectations, and concerns and incorporates eight core domains of wellbeing which are emotional wellbeing, interpersonal relations, material wellbeing, personal development, physical wellbeing, self-determination, social inclusion, and human rights [25,26]. Many autistic individuals experience poor QoL in early adulthood as they remain without appropriate support services, are employed in low paying menial jobs, and are dependent on their families, the state medical, and welfare system [27,28]. Several studies highlight factors associated with positive outcomes for autistic young adults which include a supportive social network, having employment, living independently, and access to support services [29,30]. Nevertheless, despite studies demonstrating positive change, many autistic young adults remain disadvantaged and are unable to integrate into society regardless of intellectual ability [31]. Therefore, there is a need to better understand the factors associated with successful and unsuccessful transition to employment, post-secondary education, independent living, and relationships in early adulthood for autistic young adults to inform program development and improve support for these individuals [32]. Given the link between positive outcomes in adulthood and overall personal wellbeing, QoL was used as the key outcome variable of interest and a proxy for successful transition in the current study.

Due to the evolving and lengthy nature of the transition process, longitudinal studies provide an opportunity to examine this process at repeated time points to examine specific changes made by individuals and determine what factors might be associated with successful or unsuccessful transition. A longitudinal design allows for the evaluation of transition experiences through observation of changes in lifestyle and environmental factors such as post-secondary education, employment, and attainment of independence over a 12-month transition

period [33]. Such approaches can assist in identifying those individuals most at risk for poor transition as well as potential targets for intervention that could assist at different intervention stages.

## Study aims

The aim of the present study was firstly to understand the transition journey over time for autistic young adults, and secondly, to examine the potential risk and protective factors associated with both successful and unsuccessful transition during this time. We did this by conducting a longitudinal case series examination with 9 autistic young adults. A case series design is particularly useful in that it allows for empirical inquiry and contextual analysis of unique features, events, and their relationships [34]. Further, as autism is experienced differently by individuals, a case series approach allows for an in-depth individual analysis of transition experiences by an individual in their real-world [34]. While the advantages of this case series are its prospective nature, allowing data to be collected over a 12-month period, this approach provides mainly descriptive data and allows only for statistical comparisons within subjects. Two research questions guided the present study. First, we were interested in how levels of social support and QoL change over a 12-month period for a group of 9 autistic young adults. Second, this study examined what factors are evident during the transition period that are related to a successful or unsuccessful transition to adulthood. This was addressed through examination of 9 case studies with two presented in detail to illustrate the complexities and challenges involved during the transition period, and to highlight how risk and protective factors influence individual outcomes.

## Method

### Design

This study implemented a longitudinal case study design to conduct an examination of the journey of 9 autistic participants during a transition period. The transition period referred to within the present study is a point within the life journey where young adults between the ages of 18 to 25 years old exit the school system and undertake emergent adult roles and responsibilities within society [9,35]. Specifically, this study included a repeated measures case series, with online assessments conducted at 2 time points at baseline and 12-months later. Assessments included self-report evaluations of transition planning and intervention received at high school, engagement in post-secondary education, access to employment, living circumstances, and social support.

### Participants

Participants were young adults diagnosed with autism living in Australia who completed the online survey between 1 April 2016 and 30 June 2016 at baseline, then a second time between 1 April 2017 and 30 June 2017 at follow-up. Inclusion criteria were that participants were autistic young adults without intellectual disability, currently experiencing transitions (e.g., to post-secondary education, employment), aged 18 to 25 years old, living in Australia, and completed the survey at both time points. The final sample for analysis comprised of 9 participants who were overwhelmingly female ($n = 7$) ranging from 19 to 25 years of age ($M = 20$, $SD = 2.22$). All 9 participants received a diagnosis of ASD with 4 participants diagnosed between the ages of 5 and 11 years and 5 participants diagnosed between the ages of 19 to 21 years. Details of the specific diagnosis and length of diagnosis is included within each of the 9 cases.

The data from all 9 cases are presented here to demonstrate overall changes in QoL and social support over the transition period. All 9 cases were examined in detail to provide case studies and identify factors associated with successful and unsuccessful transition. Given that we were interested in identifying risk and protective factors associated with successful and unsuccessful transition, we wanted to select cases who showed positive change in QoL and cases that showed negative change in QoL. That is, we wanted contrasting cases. Out of the 9 cases, 2 cases were identified who showed the lowest and highest QoL scores at time 1. Both also showed increases or decreases in their scores at the second assessment time point. Therefore, variability in their scores allowed examination of factors associated with positive and negative change and made them good candidates for further in-depth exploration. Both participants were male, Mr. Keith aged 23 years old who showed reliable improvement in QoL, and Mr. Reggie aged 24 years old who showed reliable deterioration in QoL. Case studies were analysed for all participants and the trends within the overall group of 9 cases are detailed below (see analytic strategy section).

## Supporting information [S1 File] 7 cases

The Supporting information [S1 File] 7 Cases contains an additional 7 cases including tables as further support and analysis of the risk and protective factors associated with transitions for autistic young adults in the study.

## Measures

An online survey was designed to measure potential indicators of successful and unsuccessful transition, as well as individual and clinical characteristics that might act as risk or protective factors to successful or unsuccessful transition outcomes. Specifically, the baseline online survey comprised of three components: (a) The author developed *About You Survey* was included to assess for demographics (e.g., age, gender, location) and individual characteristics (e.g., age at diagnosis, receipt of transition planning); (b) The Quality of Life Questionnaire [36,37] to assess QoL as a proxy for successful transitions; and (c) The Multidimensional Scale of Perceived Social Support [38] to assess perceived social support as a possible protective factor.

**About you survey.** The author developed *About You Survey* was used to collect baseline demographic information including age and identified gender. Employment status was measured by asking participants if they were in paid employment, in tertiary study, or unemployed. Living arrangements was measured by asking participants if they were living with a partner, roommate, or parents. Relationship status was measured by asking participants if they were in a relationship, single, or married. Individual characteristics were measured by asking participants age of diagnosis and whether intervention was received in behaviour support, social skills training, life skills training, and independent living skills. Participants also answered questions on transition planning at school, and whether there was individual and family involvement in transition planning, as well as whether participants received work experience placements at school and disability support. All participants completed the survey comprising 36 questions.

In order to establish credibility and trustworthiness with participants, the survey was peer reviewed by a group of four autistic young adults, who were also peer researchers at Autism Co-operative Research Centre for Living with Autism [39,40]. Feedback from the peer review process on the author developed *About You* component of the survey suggested gathering further information on participants' daily occupation. Specifically, this led to the inclusion of an additional section asking about the nature and quality of daily activities conducted by the participant (see sections 1.18 Daily activities, question 1.19 'What is your daily activity', and

question 1.20 'Please give a brief description of activities you engage in whilst at home' in the survey). In addition, the online survey was piloted with 2 autistic young adults. The 2 pilot group participants were asked to complete the draft survey and provide feedback on the visual layout of the questions and response format, wording of questions, and clarity of instructions. Feedback from the pilot review was positive with minor suggestions in the wording of 3 questions. Further, feedback from one community support organisation, prior to advertising on their website, prompted the inclusion of 4 open-ended questions in the *About You Survey* which allowed participants to provide a text response as described below.

1. Can you describe how you feel about your life at this point in time?

2. Can you describe how your daily activities/employment make you feel?

3. Can you describe the extent to which you are able to function independently on a daily basis, for example, making your own meals, getting yourself to work?

4. How would you describe your involvement socially with friends, family, and the community?

At follow-up, the same online survey was administered. In order to gain individual insights into transition experiences over a 12-month time period, 2 open-ended qualitative questions were included to provide additional information about potential changes in QoL. These were not used to measure QoL, rather to inform the case descriptions and identification of factors related to positive experiences and challenges faced during the transition period. These were as follows:

1. Over the last year what were some of the positive experiences you have had?

2. Over the last year what were some of the main challenges you have experienced?

**The quality of life questionnaire.** The QoLQ [36] is a 40-item self-report scale designed to measure the QoL of autistic individuals and those with a disability. The QoLQ consists of 4 subscales: (a) Satisfaction (SAT) as a measure of overall wellbeing with life; (b) Competence and Productivity (CP) as a measure of skills and experiences associated with access to employment; (c) Empowerment and Independence (EI) as a measure of functional independence in daily living skills; and (d) Social Belonging and Community Integration (SB) as a measure of community integration. Each subscale has 10 items that are rated on a 3-point likert scale from 1 (low) to 3 (high). Each subscale has a potential total score ranging from 10 to 30 with higher subscale scores representing higher levels of satisfaction, competence and productivity, empowerment and independence, social belonging, and overall higher QoL. Total QoL is computed by adding the 4 subscale scores with an overall potential range of 40 to 120 [37].

The instrument possesses good psychometric properties. The authors [37] report coefficient alphas for the total QoL score as .90 and for each subscale as follows: (a) Satisfaction .78; (b) Competence/Productivity .90; (c) Empowerment/Independence .82; and (d) Social Belonging .67. Test-retest reliability for the total QoL score has been reported as .87 and for each subscale as follows: (a) Satisfaction .80; (b) Competence/Productivity .96; (c) Empowerment/Independence .83; and (d) Social Belonging .82 [37].

There are no published clinical cut-offs for the QoLQ. However, individuals with disability in semi-independent or independent living, engaged in employment, with increased community integration, with a total QoL score of 80 and above are identified as having a high QoL [37]. Those individuals with disability in supervised accommodation, unemployed, with low levels of satisfaction, and a total QoL score of 79 and below are identified as having a low QoL [37]. In order to examine differences during the transition period for participants who showed

deterioration and improvement in levels of QoL, a median split of the sample based on total QoL scores was conducted at follow-up. Individuals with a total QoL score of 80 and above are identified as having a high QoL, and those with a total QoL score of 79 and below are identified as having a low QoL [37]. Subscale scores of 22 and above would likely reflect higher levels of Satisfaction, higher Competence/Productivity, greater Empowerment/Independence, and greater Social Belonging, whilst subscale scores of 21 and below would likely reflect lower levels of Satisfaction, lower Competence/Productivity, less Empowerment/Independence, and, less Social Belonging [36]. The QoLQ was used as a measure in this study as it has been validated for individuals with intellectual and developmental disabilities in Australia [32], the US, and other countries [41–44].

**The multidimensional scale of perceived social support.** The MSPSS is a 12 item self-report scale designed to measure perceived adequacy of social support from family, friends, and significant others [38]. The MSPSS has 3 subscales with 4 items per subscale corresponding with Support from Significant Other (SSO), Support from Family (SF), and Support from Friends (SFr). Participants are asked to rate items on a 7-point Likert-type scale ranging from very strongly disagree (1) to very strongly agree (7). Total subscale scores are calculated by adding responses to each of the 4 items, then calculating the average score for each subscale. The MSPSS total score is calculated by adding subscale total scores, then computing the average score. Mean scale scores ranging from 1 to 2.9 are considered low support, a score of 3 to 5 is considered moderate support, and a score from 5.1 to 7 is considered high support [26]. Higher total scores indicate increased perceptions of social support.

The psychometric properties of the MSPSS have been established. The authors [38] report Cronbach's alpha for the total MSPSS score as .88 and for each subscale as follows: (a) Support from Significant Other .91; (b) Support from Family .87; and (c) Support from Friends .85. Test-retest reliability for the total MSPSS score is reported as .85 and for each subscale as follows: (a) Support from Significant Other .72; (b) Support from Family .85; and (c) Support from Friends .75. The MSPSS has been used in research with mothers of autistic children [44] and with autistic adult populations [45,46].

## Procedure

The study was approved by the Human Research Ethics Committee (H16REA039). A letter of invitation including an information sheet about the study was e-mailed to autism support organisations and online forums. Informed consent to participate in the survey at both time points was tacit. That is, participants were asked to read the information sheet and tick a consent box in the information sheet in order to proceed. Tacit consent was assumed by subsequent progression, completion, and submission of the survey. Participants were asked to confirm that they were aged between 18 to 25 years, had a diagnosis of autism without intellectual disability, were in the transition period, and were living in Australia. Participants who met the selection criteria were eligible to participate at both time points of the survey and were asked to provide an e-mail contact address at time 1 (T1) for participation in the survey 12-months later at time 2 (T2). Once the survey was completed, the online responses were saved to a secure server that required security password access, until the completion of data analysis. Participants were provided with a pseudonym to provide anonymity in the final study write up.

At baseline a total of 16 participants, 10 who were female and 6 who were male, completed the online survey and gave consent to participate in the survey at both time points. Three male participants were excluded due to being outside of the inclusion age criteria of 18 to 25 years old. As a result, 13 participants were eligible to participate in the survey at baseline and follow-

up as part of the larger study. At follow-up, a total of 10 participants with 7 females and 3 males responded and completed the assessment Therefore 3 participants were lost to attrition. No significant differences were identified in the 3 participants lost to attrition to the 10 who completed the follow-up assessment. Further, one of the 10 participants being male, who completed the follow-up assessment was excluded from analysis as full data for the QoLQ and MSPSS was missing in their entirety. The final sample for analysis comprised of 9 participants, 7 who were female and 2 who were male, with data at both time points. Fig 1 provides a visual representation of the selection process for the inclusion and exclusion of data, and the final sample in the larger study.

## Analytic strategy

**Change in social support and QoL.** To address the first research question and to examine whether QoL and social support changed over the 12-month transition period for the sample of autistic young adults, the data was analysed in several ways. First, changes in subscale and total scores were descriptively examined at baseline and follow-up time points to determine whether measured QoL and MSPSS changed over this period. Second, the Reliable Change Index (RCI) was utilised to determine if individual participants showed reliable change on QoL from baseline to follow-up [47].

RCI's provide a measure of statistical significance regarding individual change in scores that takes into account the scale reliability and is beneficial with individual participants or small samples. Positive RCI's reflect increases and negative RCI's reflect decreases in the target score, and an RCI with a magnitude of 1.96 or greater in either direction is considered statistically reliable at the $p < .05$ level [47]. RCI's were calculated for all participants for total QoL and the 4 subscale scores to assess for statistically reliable change longitudinally from baseline to follow-up. Each participant's RCI was then categorised into, showed statistically reliable improvement, showed statistically reliable deterioration, or showed no reliable change categories to enable reporting of cases showing each type of change during the follow-up period. Individual cases were then categorised into reliable change categories and grouped according to those who showed statistically reliable improvement or statistically reliable deterioration at follow-up in terms of QoL, the primary outcome variable.

**Individual case analysis and factors associated with transition.** To address the second research question and examine risk and protective factors that were evident and potentially related to QoL during transition, a series of steps were taken. First, we conducted an inductive content analysis to examine the risk and protective factors associated with QoL [48]. Survey data was grouped into categories of factors contributing to improvement in QoL and those contributing to deterioration in QoL as reported by participants. This analysis was achieved by an in-depth coding process of each participant's response to the survey questions. The second author conducted the analysis independently to reach consensus on identified risk and protective factors [48]. Factors identified included important needs met in education, work, home, and community, as well as receipt of support services, intervention, and dependence on families and medical systems. Cases were then organised within the QoL change categories as described above (reliable improvement, reliable deterioration).

Two contrasting case studies are presented to provide further in-depth qualitative insight into how these factors were potentially associated with improvements and those with deterioration during the transition period. The process for case selection is described in the Participants section, however the remaining cases are also presented in the Supporting Information [S1 File] 7 Cases, according to reliable change category. Each case study is described in terms of demographic and individual factors, education, support, intervention, QoL, positive

TRANSITION TO ADULTHOOD

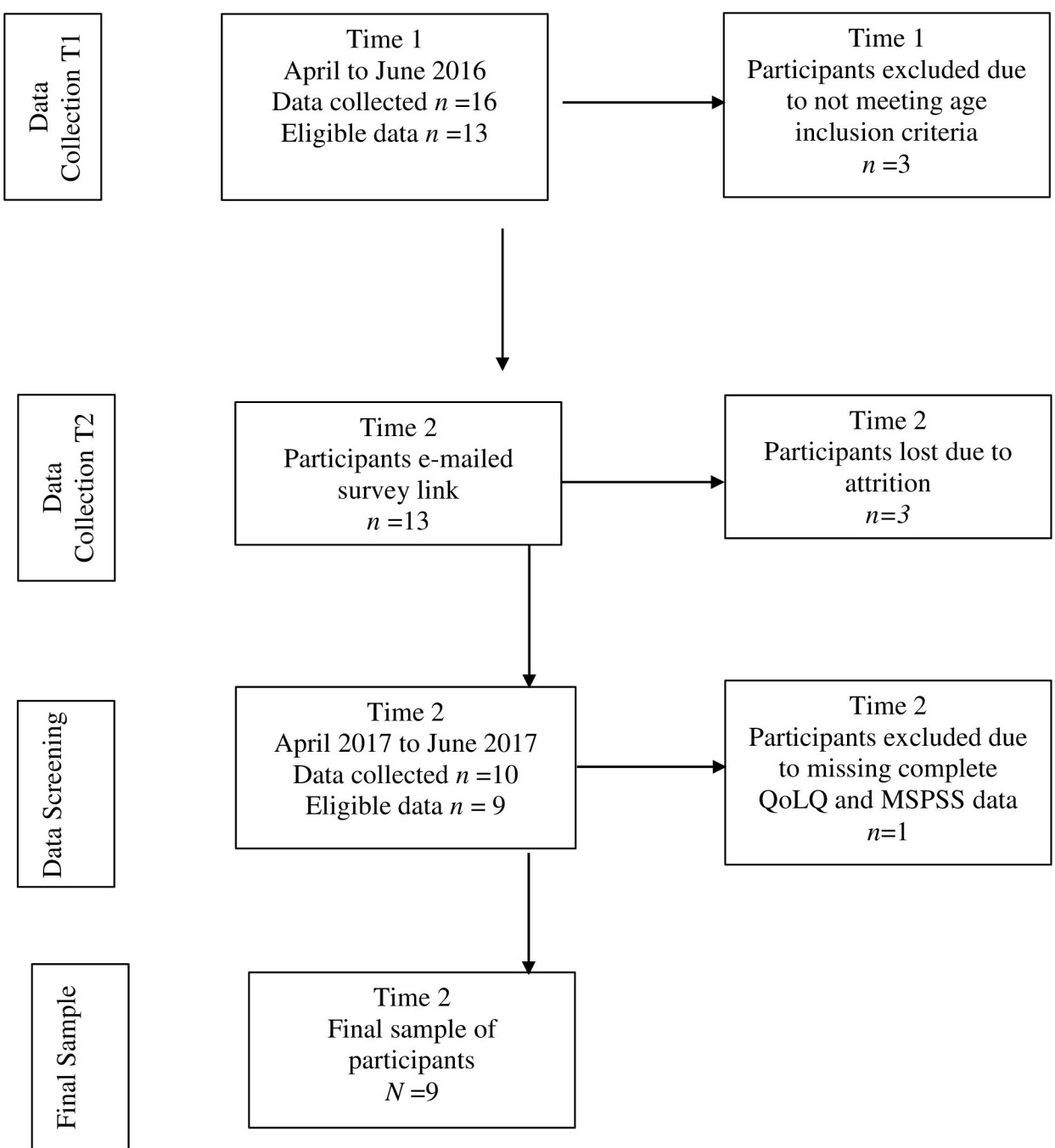

**Fig 1. Flowchart of data collection process at Time 1 (T1) and Time 2 (T2).**

experiences, social support, and autism impact and how these changed over a 12-month time period using RCI's for the key variables of QoL and social support. Descriptive data was supplemented with participant written responses to open-ended questions within the surveys at follow-up. Relevant quotes were extracted from participant responses to support their reported experiences. Common themes across cases were then summarised and contrasted.

# Results

## Demographic and individual factors

The final sample comprised 9 participants, 7 (77.8%) who were female and 2 (22.2%) who were male. The majority of participants (55.6%) lived with parents in the family home and were single. Results show that one (11.1%) lived with a partner, 3 (33.3%) lived with friends or roommates, and 4 (44.4%) were in a relationship. In addition, 5 (55.6%) received a diagnosis between the ages of 5 to 11 years whilst 4 (44.4%) were between 19 to 21 years of age, and 6 (66.7%) received fortnightly disability pension payments and 3 (33.3%) did not. Responses show that 5 (55.6%) were engaged in post-secondary education, 2 (22.2%) in paid employment, one (11.1%) stayed at home, and one (11.1%) did not report. Weekly wages for the 2 participants (22.2%) who were in employment was over $200 per week, with average hours worked between 20 to 30 hours per week. A detailed summary of baseline demographic information can be found in Table 1.

## Changes in social support and QoL

Participants were grouped according to their RCI category of change, that is, those who improved in QoL from baseline to follow-up (Improvement in Quality of Life), those who deteriorated from baseline to follow-up (Deterioration in Quality of Life), and those who showed no change in QoL from baseline to follow-up (No Change in Quality of Life). An equal number of participants 4 (44.4%) showed improvement in QoL as did those showing deterioration. No change in QoL was evident for only one participant (11.1%). Six (66%) participants received moderate social support, 2 (22%) participants received high social support and one (11%) participant demonstrated an increase from moderate to high social support over the transition period. QoL groupings were subsequently utilised in the following case series analysis, with participant results discussed according to their QoL status and social support over the 12-month follow-up period. A summary of these findings is presented in Table 2 in addition to descriptions of the level of QoL at baseline and follow-up.

For the Satisfaction subscale, clinically reliable improvement was evident for 4 (44.4%) participants with the remaining participants showing reliable deterioration. For the Competence/ Productivity subscale, clinically reliable improvement was evident for 4 (44.4%) of participants, with 4 (44.4%) participants showing no change and one (11.1%) participant showing reliable deterioration. For the Empowerment/Independence subscale, clinically reliable improvement was only evident for 2 (22.2%) participants, with the majority, (66.6%) showing no reliable change. Finally, for the Social Belonging subscale, equal numbers of participants (one third) showed reliable improvement, reliable deterioration, and no change. In sum, 4 participants showed improvement on total QoL and the Satisfaction subscale, whilst 3 out of the same 4 participants showed improvement on the Empowerment and Independence, and Social Belonging subscales, and half showing improvement on the Competence and Productivity subscale. Therefore, those participants who showed improvement in total QoL also tended to improve in Empowerment and Independence, Competence and Productivity, overall Satisfaction, and Social Belonging. A summary of these findings is presented in Tables 3 and 4.

Table 1. Participant demographics.

| Category | n | % |
|---|---|---|
| **Gender [Age *M* = 21]** | | |
| Male | 2 | 22.2 |
| Female | 7 | 77.8 |
| **Age of Diagnosis** | | |
| 4 to 13 years | 5 | 55.6 |
| 19 to 21 years | 4 | 44.4 |
| **Living Circumstances** | | |
| Partner | 1 | 11.1 |
| Friends/roommates | 3 | 33.3 |
| Parents | 5 | 55.6 |
| **Relationship Status** | | |
| Relationship | 4 | 44.4 |
| Married | - | - |
| Single | 5 | 55.6 |
| **Disability Pension** | | |
| Yes | 6 | 66.7 |
| No | 3 | 33.3 |
| **Current Post-Secondary Education** | | |
| Yes | 5 | 55.6 |
| No | 3 | 33.3 |
| Missing | 1 | 11.1 |
| **Program** | | |
| University Degree | 4 | 44.4 |
| Technical and Further Education | 1 | 11.1 |
| None | 4 | 22.2 |
| **Paid Employment at Baseline** | | |
| Yes | 2 | 22.2 |
| No | 7 | 77.8 |
| **Job Category** | | |
| Vocational Trade | 1 | 11.1 |
| Administrative | 1 | 11.1 |
| No job category | 7 | 77.8 |
| **Weekly Pay** | | |
| Over $200 | 2 | 22.2 |
| No weekly pay | 7 | 77.8 |
| **Average hours worked** | | |
| 20–30 | 2 | 22.2 |
| None | 7 | 77.8 |

## Factors associated with transition

There was a great deal of similarity in protective factors identified across cases, though greater variation in the risk factors identified for poorer transition. Social support in particular, appeared as a common protective factor for those participants who were unemployed during transition. With regards to risk factors associated with poorer transition, functional independence and unemployment were evident for many as was mental health problems, however, the type of mental health problem differed between participants. Table 5 presents a summary of

**Table 2. RCI's for quality of life total scores (*N* = 9).**

| Case | QoLT-T1 | QoLT-T2 | RCI | Category | QoL-T1 High/Low | QoLT-T2 High/Low | MSPSS-T1 High/Moderate/Low | MSPSS-T2 High/Moderate/Low |
|------|---------|---------|-----|----------|-----------------|------------------|----------------------------|----------------------------|
| Ms. Katherine | 63 | 87 | +19.09 | Reliable improvement | Low QoL | High QoL | Moderate | Moderate |
| Ms. Kelly | 69 | 66 | - 2.38 | Reliable deterioration | Low QoL | Low QoL | Moderate | High |
| Ms. Kylie | 85 | 78 | -5.57 | Reliable deterioration | High QoL | Low QoL | Moderate | Moderate |
| Mr. Keith | 88 | 96 | +6.36 | Reliable improvement | High QoL | High QoL | Moderate | Moderate |
| Ms. Lily | 79 | 80 | +0.79 | No reliable change | Low QoL | Low QoL | Moderate | Moderate |
| Ms. Lavender | 75 | 64 | -8.75 | Reliable deterioration | Low QoL | Low QoL | Moderate | Moderate |
| Ms. Petal | 72 | 82 | +7.95 | Reliable improvement | Low QoL | High QoL | Moderate | High |
| Mr. Reggie | 79 | 69 | -7.95 | Reliable deterioration | Low QoL | Low QoL | High | High |
| Ms. Talita | 79 | 99 | +15.91 | Reliable improvement | Low QoL | High QoL | Moderate | Moderate |

RCI = reliable change index values of 1.96 or greater in either direction indicate a reliable change at the 5% significance level or better.

the risk and protective factors associated with successful and unsuccessful transition, identified through the case study analysis for all 9 cases.

## Individual case analysis

Two case illustrations are presented below in detail to further illustrate how the risk and protective factors described in Table 3 influenced QoL during the transition process. The 2 case studies were drawn from the 2 categories of showing 'reliable improvement in QoL' and 'reliable deterioration in QoL' in order to differentiate between those participants who transitioned successfully and those who did not.

### Mr Keith: Improvement in quality of life

**Demographic characteristics.** Mr. Keith is a 23-year-old single male who lives with his parents, is employed as a kitchen hand for 20 to 30 hours a week, earns an income of $200 per week, and receives the disability pension. Mr. Keith received an autism diagnosis at the age of 5. He therefore has had his diagnosis for a period of 18 years.

**Education, support, and intervention.** Mr. Keith attended a state high school, received support through the special education program, and completed year 12. At school Mr. Keith received a transition plan with parent involvement in transition planning, and received the

**Table 3. RCI's for Quality of Life Subscale Scores Satisfaction (SAT) and Competence and Productivity (CP) (*N* = 9).**

| Case | SAT T1 | SAT T2 | RCI | Category | CP T1 | CP T2 | RCI | Category |
|------|--------|--------|-----|----------|-------|-------|-----|----------|
| Ms. Katherine | 13 | 18 | +3.97 | Reliable improvement | 11 | 24 | +10.34 | Reliable improvement |
| Ms. Kelly | 25 | 19 | -4.77 | Reliable deterioration | 10 | 15 | +3.97 | Reliable improvement |
| Ms. Kylie | 27 | 24 | -2.38 | Reliable deterioration | 15 | 14 | -0.79 | No reliable change |
| Mr. Keith | 24 | 30 | +4.77 | Reliable improvement | 23 | 22 | -0.79 | No reliable change |
| Ms. Lily | 20 | 16 | -3.18 | Reliable deterioration | 12 | 23 | +8.75 | Reliable improvement |
| Ms. Lavender | 20 | 13 | -5.57 | Reliable deterioration | 20 | 20 | 0 | No reliable change |
| Ms. Petal | 15 | 18 | +2.38 | Reliable improvement | 19 | 18 | -0.79 | No reliable change |
| Mr. Reggie | 18 | 14 | -3.18 | Reliable deterioration | 20 | 15 | - 3.97 | Reliable deterioration |
| Ms. Talita | 15 | 25 | +7.95 | Reliable improvement | 21 | 28 | +5.57 | Reliable improvement |

*Note.* RCI = reliable change index values of 1.96 or greater in either direction indicate a reliable change at the 5% significance level or better.

**Table 4. RCI's for Quality of Life Subscale Scores Empowerment Independence (EI) and Social Belonging (SB) (N = 9).**

| Case | EI T1 | EI T2 | RCI | Category | SB T1 | SB T2 | RCI | Category |
|------|-------|-------|-----|----------|-------|-------|-----|----------|
| Ms. Katherine | 22 | 25 | +2.38 | Reliable improvement | 17 | 20 | +2.38 | Reliable improvement |
| Ms. Kelly | 12 | 13 | +0.79 | No reliable change | 22 | 19 | -2.38 | Reliable deterioration |
| Ms. Kylie | 18 | 17 | -0.79 | No reliable change | 25 | 23 | -1.59 | No reliable change |
| Mr. Keith | 20 | 20 | 0 | No reliable change | 21 | 24 | +2.38 | Reliable improvement |
| Ms. Lily | 23 | 23 | 0 | No reliable change | 24 | 17 | -5.57 | Reliable deterioration |
| Ms. Lavender | 17 | 16 | -0.79 | No reliable change | 18 | 15 | -2.38 | Reliable deterioration |
| Ms. Petal | 21 | 25 | +3.18 | Reliable improvement | 17 | 21 | +3.18 | Reliable improvement |
| Mr. Reggie | 23 | 20 | -2.38 | Reliable deterioration | 18 | 20 | +1.59 | No reliable change |
| Ms. Talita | 23 | 25 | +1.59 | No reliable change | 20 | 21 | +0.79 | No reliable change |

*Note.* RCI = reliable change index values of 1.96 or greater in either direction indicate a reliable change at the 5% significance level or better.

following interventions: behaviour support, social skills training, independent living skills, and life skills training. Additionally, he received support through disability employment services and had two work experience placements whilst at school. Mr. Keith has a Technical and Further Education College (TAFE) qualification and reported having received life skills training and independent living skills training post high school. Thus, Mr. Keith has completed his education, obtained additional post-secondary education qualifications, and specific skills to enhance his transition.

**Quality of life.** Mr. Keith showed clinically reliable improvement in scores on the total QoL, Satisfaction, and Social Belonging subscales from baseline to follow-up. Scores on the Competence/Productivity remained consistently high from baseline to follow-up, thus indicating that he is confident with the skills and experience required for his employment, whilst scores on the Empowerment/Independence subscale remained low for Mr. Keith over time. Notably, Mr. Keith did report receiving comprehensive support at baseline, which may have contributed to his high QoL score at baseline. RCI scores for total QoL and subscales from baseline to follow-up are presented in Table 6.

**Table 5. Risk and protective factors (N = 9).**

| Case | Risk Factors | Protective Factors |
|------|-------------|-------------------|
| Ms. Katherine | Unemployed | Social support |
| Ms. Kelly | Unemployed<br>Anxiety | Social support |
| Ms. Kylie | Unemployed<br>Functional Independence | Social support |
| Mr. Keith | Functional Independence | Employment<br>Training<br>Social support |
| Ms. Lily | Unemployed | Social Support |
| Ms. Lavender | Unemployed<br>Depression<br>Anxiety | Social Support |
| Ms. Petal | Unemployed | Social Support |
| Mr. Reggie | Unemployed<br>Functional Independence | Social support |
| Ms. Talita | Stress<br>Anxiety | Employment<br>Training<br>Social support |

**Table 6. Summary of QoLQ, RCI's, and MSPSS categories for Mr. Keith.**

|  | T1 | T2 | RCI | Category |
|---|---|---|---|---|
| Quality of Life Total | 88 | 96 | +6.36 | Reliable improvement |
| Satisfaction | 24 | 30 | +4.77 | Reliable improvement |
| Competence/Productivity | 23 | 22 | -0.79 | No reliable change |
| Empowerment/Independence | 20 | 20 | 0 | No reliable change |
| Social Belonging | 21 | 24 | +2.38 | Reliable improvement |
|  | **T1** | **T2** | **T1** | **T2** |
| Multidimensional Scale of Perceived Social Support Total | 4 | 7 | Moderate | High |
| Support from Significant Other | 2 | 7 | Low | High |
| Support from Family | 7 | 7 | High | High |
| Support from Friends | 4 | 6 | Moderate | High |

MSPSS scores from 1 to 2.9 indicate low support, 3 to 5 moderate support, and 5.1 to 7 high support.

**Social support.** Perceived social support scores show an increase on the total Multidimensional Scale of Perceived Social Support score, Support from Significant Other, and Support from Friends subscales over time, whilst Support from Family for Mr. Keith remained high at both time points. Specifically, at 12-month follow-up, Mr. Keith reported that he enjoyed interacting with friends and family on a daily basis and at work he enjoyed meeting customers. In his words, 'I participate fully with friends and family'. Thus, social support appeared to be a particularly positive feature in Mr. Keith's transition, especially support from his family.

**Challenges.** Scores on the Empowerment/Independence subscale showed no clinically reliable change for Mr. Keith over time, thus indicating Mr. Keith's independence in daily living activities remained low. Mr. Keith reported no major challenges; however, he did note feelings of inadequacy in not being able to cook meals for himself. Further, Mr. Keith reported difficulty in accessing transport to visit his Grandmother at the nursing home, which was important to him. In addition, he wished that he could attend more community events, and specifically mentioned music concerts. Thus, it would appear that access to transport and mobility may present some difficulty in achieving functional independence for Mr. Keith.

**Positive experiences.** At follow-up, scores on the Satisfaction and Social Belonging subscales showed clinically reliable improvement for Mr. Keith. Indeed, Mr. Keith reported being a volunteer at a day-care centre for children with disabilities. In his words, 'I feel a sense of fulfilment in helping kids, who are having difficulties understanding why they are different because of the ASD'. Thus, it appeared that Mr. Keith experienced meaningful and positive experiences by interacting with and supporting autistic children during the follow-up period.

**Autism impact.** For Mr. Keith whilst he reported improvement in QoL overall, and several positive experiences, there were still notable ways in which his ASD impacted his life. Specifically, Mr. Keith reported difficulty accessing transport and attending community events. However, in particular, Mr. Keith reported being happy with his life, satisfied with his work and in his words, he reported that he, 'Enjoys doing the same things every day'. Thus, maintaining a routine was important and beneficial to Mr. Keith's overall QoL.

## Mr Reggie: Deterioration in quality of life

**Demographic characteristics.** Mr. Reggie is a 24-year-old-male, is in a relationship, is unemployed, and lives with his parents. He is engaged in part-time study at university and receives the disability pension. Mr. Reggie's daily activity includes online gaming, online social

interaction with his girlfriend, and sleeping. Mr. Reggie received an autism diagnosis at age 11. He therefore has had his diagnosis for a period of 13 years.

**Education, support, and intervention.** Mr. Reggie attended a state high school and completed Year 12. At high school, Mr. Reggie did not receive support or interventions, did not access work experience placements, and did not receive a transition plan.

**Quality of life.** Mr. Reggie showed clinically reliable deterioration in scores on the total QoL, Satisfaction, Competence/Productivity, and Empowerment/Independence subscales from baseline to follow-up. Scores on the Social Belonging subscale remained low for Mr. Reggie at both baseline and follow-up, thus indicating ongoing difficulty in community integration throughout the transition period. RCI scores for total QoL and subscales from baseline to follow-up are presented in Table 7.

**Social support.** Whilst Mr. Reggie's reported perceived social support scores remained high on the Support from Significant Other and Support from Family subscales over time, and his perceived Support from Friends remained moderate, with notable difficulties in this area noted. Specifically, at 12-month follow-up, Mr. Reggie reported that his girlfriend, a key source of support, moved interstate, and that he lost contact with his friends over the follow-up period, due to his inability to socialise with them. In his words he reported that, 'I will go out with friends about two times a year' indicating that he did not access support from friends regularly. In contrast, Mr. Reggie reported that his family provided him with encouragement by helping him socialise. For example, he stated, 'They push me do things that I don't like to do, but I have to, like going to my Grandma's birthday.' Thus, support from family appeared to be encouraging for Mr. Reggie.

**Challenges.** Low scores on the Satisfaction, Competence/Productivity, and Social Belonging subscales indicated that overall, Mr. Reggie was unhappy with his life situation, and experienced difficulty accessing employment. Specifically, he reported feeling incompetent in enrolling at university and seeking employment. Indeed, Mr. Reggie reported major challenges relating to study and employment. Further, Mr. Reggie stated that, 'I want to move forward but find it difficult to engage with people and businesses.' It may also be that Mr. Reggie's mental health challenges could be contributing to his difficulties with socialising and finding employment. In his words, Mr. Reggie also reported that he was, 'depressed and unmotivated' which were additional challenges for him during the transition period. Thus, Mr. Reggie's challenges in communicating and interacting with people, and feeling unproductive in his life appeared to impact his overall mental health and wellbeing during the transition process.

**Table 7. Summary of QoLQ, RCI, and MSPSS categories for Mr. Reggie.**

|  | T1 | T2 | RCI | Category |
|---|---|---|---|---|
| Quality of Life Total | 79 | 69 | -7.95 | Reliable deterioration |
| Satisfaction | 18 | 14 | -3.18 | Reliable deterioration |
| Competence/Productivity | 20 | 15 | -3.97 | Reliable deterioration |
| Empowerment/Independence | 23 | 20 | -2.38 | Reliable deterioration |
| Social Belonging | 18 | 20 | +1.59 | No reliable change |
|  | T1 | T2 | T1 | T2 |
| Multidimensional Scale of Perceived Social Support Total | 6 | 6 | High | High |
| Support from Significant Other | 7 | 6 | High | High |
| Support from Family | 7 | 7 | High | High |
| Support from Friends | 5 | 4 | Moderate | Moderate |

MSPSS scores from 1 to 2.9 indicate low support, 3 to 5 moderate support, and 5.1 to 7 high support.

**Positive experiences.** Mr. Reggie was able to identify some positive experiences during the 12 months. For example, he reported getting his driver's licence as a positive experience, however in his words he also reported, 'I don't like going out, so I get mum to do errands for me.' Thus, while he achieved something positive, he was unable to integrate this fully into his daily routine. As he was able to book flights and accommodation to visit his partner interstate, Mr. Reggie viewed these abilities as a positive experience.

**Autism impact.** For Mr. Reggie, the impacts of his autism and associated challenges appeared to have a considerable effect on his daily activities. Overall, Mr. Reggie reported ongoing feelings of inadequacy in communicating with people. Notably, communication difficulties are a hallmark of autism [49]. Thus, communication difficulties remained a challenge for him. Further, Mr. Reggie appeared to experience some of the negative stigma attached to autism during the transition period, as he reported, 'People see the diagnosis and think I can't do things.' He reported that this affected his behaviour, and often meant that he asked his mother to contact people at places of importance on his behalf. For example, he reported, 'I get mum to contact places, like the university for enrolment, but then people treat me weird after.' Thus, it would appear that in this case, family advocacy in communicating for Mr. Reggie as a young adult, presented a barrier for him in later social interactions.

## Discussion

This study examined the transition experiences of a target group of participants over a 12-month period to evaluate potential risk and protective factors associated with successful or unsuccessful transition during this time. Nine cases illustrated challenges and complexities during this period and 2 were presented in further detail to represent those who demonstrated improved QoL and those who deteriorated in QoL from baseline to follow-up and to demonstrate how risk and protective factors might influence transition. The final number of participants represented in the study were 2 males and 7 females. Whilst females appear to be over-represented in the study, there is ongoing discussion in the literature of the increase in prevalence estimates of autism in females [4].

Overall, at baseline, only a few participants showed high QoL, with some showing high satisfaction with life, high functional independence, and high social belonging. At baseline, most participants showed low QoL and experienced low levels of satisfaction, low confidence and productivity in skills and experience required to access employment, low levels of functional independence in daily activities, and low levels of social belonging. Therefore, at a crucial point in their lives, most of the young adults in this sample are attempting to navigate challenges associated with transitions with a relatively low starting point of confidence and skills. These findings support previous research with autistic individuals entering young adulthood, where the majority reported low QoL, were dependent on family, required support in daily activities, with limited social interaction [6,7]. Thus, these findings highlight that many autistic individuals transitioning to adulthood experience overall poor social and psychological wellbeing and concurs with previous research [32].

Overall, from baseline to follow-up, the young adult group as a whole showed some improvement on competence and productivity in access to employment, independence in daily activities, and total QoL, whilst overall satisfaction with life, integration with the community, and social belonging remained consistently low. Autism specific planning and interventions, skills development, parental involvement in transition planning, work experience placements, and support from service providers appeared to be protective factors for Mr. Keith and all participants who experienced improved QoL and a successful transition. These findings are consistent with recent research which indicates with focused skills training, access

to work experience placements, as well as parental involvement, autistic young adults can be successful in transitioning to post-secondary education, employment, and experience an improved QoL [12,31].

A lack of engagement in post-secondary education and unemployment were associated with deterioration in QoL for Mr. Reggie and all cases who experienced an unsuccessful transition. Further, lack of interventions in high school with no parental involvement, or support from a disability service provider, with no transition focus on educational and functional outcomes, had a negative impact on successful engagement in post-secondary education and employment. This finding provides further support for the importance of targeted transition planning, parental and interagency support as discussed in previous research [16,17]. Co-occurring depression and anxiety, limited social skills, and communication challenges appeared to be risk factors for unsuccessful transition. It is therefore imperative that disability support services are equipped to facilitate mental health intervention for these individuals.

Outcomes across all 9 cases suggest that social support from family was an important protective factor for this group, whilst lower independence appeared to be a risk factor for both successful and unsuccessful transition. Quite unexpectedly, parental support was also perceived as a potential barrier for Mr. Reggie who experienced unsuccessful transition. One possible explanation for this, is that parents of autistic individuals can be perceived as overly protective, likely due to the perception that their child is incapable of self-managing challenges associated with the transition process [21]. As such, in the present study, the appearance of Mr. Reggie being a young adult attending university and the associated social expectation of being independent, combined with having a parent advocate for him at enrolment, presented a confusing situation for university staff. This finding helps to illustrate the point that regardless of intellectual ability autistic young adults still experience difficulties engaging in the community [31]. Interestingly, the importance of an autism diagnosis and associated challenges contributed to identity formation, self-awareness, and self-efficacy, potentially as a protective factor for successful transition, in navigating and managing transition challenges for both Mr. Keith and Ms. Katherine.

Overall, across all cases, consistent with current research, social and communication difficulties, co-occurring depression and anxiety, and challenges in adaptive behaviour associated with autism emerged as risk factors to a successful transition to post-secondary education, employment, independent living, and friendship formation [23]. Support from friends, families, and significant others appeared to be protective factors and provides further support for research in this area [29–31]. Further, employment and earning a wage increased self-esteem and led to success in adulthood.

## Implications of study findings

Study findings show that young adults appeared to be limited not by their individual difficulties, but by the very systems charged with supporting them through their schooling, post high school activities, and transition to key adult roles within the community. Although some autistic young adults showed success in transition and improvements in QoL over the 12-month transition period, all young adults in this study noted some level of difficulty with skills required to access paid employment, challenges in social and community integration, and organisational skills in daily living activities.

Challenges associated with poor outcomes for young adults during transitions were due to first, insufficient professional attention to their abilities at the school and post school systemic level, second, limited knowledge of the developmental nature of autism, and third, limited understanding of the implementation of individualised interventions that will facilitate

successful outcomes in this population, particularly as they transition to adulthood. The implication of these findings suggests a comprehensive understanding of the nature of autism as a lifelong developmental condition by all individuals who interact with the young autistic adults, including family, teachers, peers, disability support staff, and workplace colleagues is crucial. Therefore, ongoing autism specific intervention both at school and post-school may be beneficial.

At the school level, intervention promoting social communication with peers, developing skills required in organisation, time management and budgeting, are specific skills that may help to foster success in the social, vocational, and post-secondary education domains during the transition period. The evidence from our study suggests that knowledge of autism specific skills intervention is important in informing autism policy development and practice within disability support infrastructure and suggests that a transition focused education is important. These findings are of critical importance to a wide audience including educational institutions at a school and post-school level, policy developers, and families.

## Strengths and limitations

The present study utilised a longitudinal case series design incorporating 2 assessment time points to determine how QoL changed over this period. The case series allowed for an in-depth contextual analysis of unique features and individual transition experiences of autistic young adults in their real-world. Such case illustrations can inform focused goal planning at the school and post-school level, to inform program development for agencies supporting these individuals, and to contribute specialised knowledge to strategic policy development. The case study methodology was rigorous, with measures used to assess changes in QoL and social support. Further, the use of RCI's provided statistical rigour to the examination of effects. Nevertheless, there are several limitations.

Given the heterogeneity of an autism diagnosis, the young adults in the present study cannot necessarily be considered representative of the broader autistic population. Despite the common risk and protective factors identified, it is difficult to understand the magnitude of effect due to the small sample size. At the time of the surveys, data on socio-economic status of the family of origin and race was not collected which may have influenced outcomes. It is important to note that despite research which suggests autism is more common in males, our sample was predominantly female. The 2 male cases presented in detail in this manuscript were deliberately chosen as they represented similar stages of transition, age, gender and clearly represented an example of deterioration and improvement. Whilst our overall group outcomes suggest similarities across the sample, and the other 7 female cases presented in the Supporting Information [S1 File] 7 Cases do provide some insights into transition processes and QoL outcomes for females compared to males, the lack of direct gender comparisons must be acknowledged as a limitation of this research. Data collection via the survey was restricted to participants who had access to a computer. Only having access to a computer is an example of selection bias. Multiple approaches to data collection could have potentially increased the sample size. In conjunction to using the online survey, distributing and collecting hard copies of surveys from participants through autism associations may have yielded more responses.

Finally, whilst a longitudinal case series design captured rich data from 9 cases, it is difficult to generalise findings from the present study given the heterogeneity of an autism diagnosis in specific behaviour traits, social skills, and cognitive ability ascribed to an individual. Future studies should examine the impacts in a large cross-sectional population and compare to the existing evidence base. Research with parents and teachers would enrich the study findings by contributing social validity.

## Conclusion

This case series highlighted specific demographic, individual, and group characteristics that facilitate a successful transition to adulthood for autistic young adults. At a systems level, this is important in informing family, disability organisations, and key stakeholders on strategic policy in tailored interventions to ensure a seamless transition to post-secondary education, employment, independence, social inclusion, and overall life satisfaction for autistic young adults.

## Supporting information

**S1 File.**
(DOCX)

## Author Contributions

**Conceptualization:** Yosheen Pillay.

**Data curation:** Yosheen Pillay, Sonja March.

**Formal analysis:** Yosheen Pillay, Sonja March.

**Investigation:** Yosheen Pillay, Sonja March.

**Methodology:** Yosheen Pillay, Sonja March.

**Project administration:** Yosheen Pillay.

**Resources:** Yosheen Pillay.

**Supervision:** Charlotte Brownlow, Sonja March.

**Writing – original draft:** Yosheen Pillay, Charlotte Brownlow, Sonja March.

**Writing – review & editing:** Yosheen Pillay, Charlotte Brownlow, Sonja March.

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
