## [Decision Letter · Decision Letter 0]

15 Feb 2022

PONE-D-21-38878Transition approaches for autistic young adults: a case series studyPLOS ONE

Dear Dr. Pillay,

Thank you for submitting your manuscript to PLOS ONE. After careful consideration, we feel that it has merit but does not fully meet PLOS ONE’s publication criteria as it currently stands. Therefore, we invite you to submit a revised version of the manuscript that addresses the points raised during the review process.

I would suggest that in revising your manuscript you pay particular attention to reviewer 3's suggestions regarding the methodology and results section. I do not think you need to add much to the introduction, but you might consider including a bit more reference to the research on experiences of transition.  Please ensure that your decision is justified on PLOS ONE’s publication criteria and not, for example, on novelty or perceived impact.

We look forward to receiving your revised manuscript.

Kind regards,

Amanda A. Webster

Academic Editor

PLOS ONE

Journal Requirements:

3. Please upload a copy of Supporting Information S1 which you refer to in your text on pages 8, 15 and 28.

Reviewers' comments:

Reviewer's Responses to Questions

**Comments to the Author**

1. Is the manuscript technically sound, and do the data support the conclusions?

Reviewer #1: Yes

Reviewer #2: Yes

Reviewer #3: Partly

2. Has the statistical analysis been performed appropriately and rigorously? 

Reviewer #1: Yes

Reviewer #2: Yes

Reviewer #3: No

3. Have the authors made all data underlying the findings in their manuscript fully available?

Reviewer #1: Yes

Reviewer #2: Yes

Reviewer #3: Yes

4. Is the manuscript presented in an intelligible fashion and written in standard English?

Reviewer #1: Yes

Reviewer #2: Yes

Reviewer #3: Yes

5. Review Comments to the Author

Reviewer #1: This paper describes an interesting study conducted ethically and with methodological rigour. It provides qualitative data that describe the impact of transition on young adults on the autism spectrum with reference to 9 specific cases.

The only area where I think you could improve is in the conclusions you draw from some of the evidence you provide. For example, in the case of Mr. Reggie you conclude that his lack of employment and social connection could have led to his mental health challenges. You do not consider whether his mental health issues could be the reason for his difficulties with socialising and finding employment. You may well have additional data that has led you to some of your conclusions but these are not always evident in the material provided in the paper.

On the whole this paper is well-written and clearly presented. Well done.

Reviewer #2: Overall this is an excellent article that has touched on a very important issue that confronts many autistics and their families, namely the transition to adult stages.

However, to improve this manuscript I advise the following four minor modifications:

1. For the reader it would be good to have a section about what ASD diagnosis each participant received (i.e., Asperger syndrome under the DSM-IV-TR) and (if the data was captured) how many years has the participant had their diagnosis.

2. A copy of the About You survey as an attachment. This will assist others to conduct a replication or repeat of the study.

3. In Table 1 “Living circumstances” and “Program” do not add up to 9 participants. Please explain or correct this numerical discrepancy.

4. In the Strengths and Limitations section please write a small paragraph stating that one limitation of the study was that it was restricted to only participants who had access to a computer. Only having access to a computer is an example of selectin bias. Also it would be appropriate to highlight that multiple approaches to data collection would have potentially increased the sample size. For example, in conjunction to using an online survey, distributing and collecting hard copies of surveys from participants who attended an event at an autism association might have yielded more responses.

Reviewer #3: Review for PLOS-One

Transition approaches for autistic young adults: a case series study

Introduction

The introduction needs to be rewritten to emphasise the importance of transition for young adults with Autism . T here is very limited reference to the transition literature in general and transition literature and for young adults with Autism . The same comment could be made about Quality of Life. The richness of the quality of life literature is missing and is needed as QoL is a key outcome variable and a proxy variable for transition.

Study Aims and Design

The study aims are clearly stated and are appropriate to longitudinal design. However, I would question whether a case series design will as the authors claim, _allow for empirical inquiry and contextual analysis of unique features, events, and their relationships.’”

I think that the authors need to rewrite this section so there is greater acknowledgement of strengths and weaknesses of Case Series studies. For instance, I question whether they provide empirical data because of the lack of a comparison group. The strengths of some of the features of this case series design such as that it is a prospectïve design and data was collected over a 12 month time period are acknowledged. However, I think the authors should acknowledge that only descriptive data and statistics should be used. Case series design studies are a good way to generate new hypotheses but, in my opinion, do not deliver high levels of empirical evidence.

Results

The demographic results are clearly presented but there is no discussion of them in the text. For example, is there any reason for the overwhelming number of participants being female?

I am confused by Page 8 line 156 which says all of the data relates to Quality of Life. Line 163 then states that there were significant increases in QOL.

The word significant needs to be replaced in this sentence as the design does not allow for statements such as this to be made.

This next paragraph then goes on to describe other measures that in line 173 they state as being indicators of successful and unsuccessful transitions.

I think these claims need to be tempered given the limitations of the design, the quality of the measures and the lack of representative sample that is being presented and the lack of inclusion of the transition literature previously mentioned under the Introduction section.

I also have to question whether two open-ended questions added to the About Me Survey can really assess changes in quality of life.

Results

There are some issues with the decisions made in this section as well which reflect lack of knowledge of quality of life measurement. The authors have defined high quality of life as above 80 and low quality of life as 79 and below.

In previous research 75 has been categorised as average quality of life across populations. I would suggest that the authors need to provide support for this decision.

The second issue is that all authors in this field recognise that Quality of Life is a multifaceted construct, yet the authors have used the total scores for analysis of statistical change.

I would suggest that the authors reanalyse this section using the subscale scores. Using the total scores could lead to poor conclusions and masking of important interactions.

Given the extensive reanalysis and rewriting that I have recommended I do not think it is necessary to comment on the final sections of the paper.

These may need to be extensively rewritten in light of the reanalysis of the Quality of Life subscales

This is an interesting topic but I feel that this paper needs major revisions and reanalysis of some data before it could be accepted for publication.

Thank you for the opportunity to review this paper

Best Wishes

6. PLOS authors have the option to publish the peer review history of their article (what does this mean?). If published, this will include your full peer review and any attached files.

Reviewer #1: **Yes: **Dr Debra Costley

Reviewer #2: No

Reviewer #3: No

---

## [Author Response · Author response to Decision Letter 0]

14 Mar 2022

Reviewer 1 Comment

 Author Response

Mr Reggie-Consider whether his mental health issues could be the reason for his difficulties socialising and finding employment

 Thank you for this suggestion. We believe this is an important consideration and have edited as such. A sentence has been added to reflect this on page 27

Reviewer 2 Comment

 Author Response

What ASD Diagnosis did each participant receive and how many years have participants had their diagnosis This information has been included on page 7. Additional information regarding age of diagnosis, length, and type of diagnosis (e.g. ASD, aspergers) has been included in each of the cases in the supplementary file submitted as an attachment as well as the two cases in the text.

Please provide a copy of the About You Survey instrument

 We are pleased to share this author developed About You Survey. This has been included in the submission

Table 1 correction of typo to add up to 9 participants under Program

 Apologies for this typo. This has been corrected in Table 1 under Program to reflect a total of 9 participants

Limitation - access to computer was a limitation to only participants who had a computer

 Thank you for highlighting this important point. We acknowledge this as a limitation and have now included this as part of our limitation section on P33. As suggested, we have also suggested that multiple approaches to data collection would have potentially increased the sample size.

Reviewer 3 Comment

 Author Response

There is limited reference to the transition literature in general and for young adults with autism

 We agree and have reviewed the literature and included additional commentary within the introduction as well as 10 new references to reflect the richness of transitions in general and specifically for autistic individuals. This is now also included in the reference list.

The richness of the quality of life literature is missing

 We agree and have reviewed the literature and included additional commentary within the introduction as well as 7 new references to reflect the richness of transitions in general and specifically for autistic individuals. This is now also included in the reference list.

Strengths and weaknesses of a case series design

 We have edited the study aims section on pages 5 and 6 to temper the description of a case series design and focus on strengths and weaknesses of this methodological approach

Results

The demographic results are clearly presented but there is no discussion of them in the text.

 Thank you for highlighting this omission. We have now added a discussion the demographic and individual factors in the results section on page 15.

Is there a reason for the overwhelming number of female participants? We acknowledge your thoughtful comments relating to gender. It certainly was interesting in this study to see our sample represented by 7 women and only 2 men, and we thank you for highlighting the need to discuss this in the paper. 

We have added in detailed information regarding the gender split at the time of data collection both at Time 1 and Time 2 which can be seen in Figure 1 and on page 12. At Time 1 the initial participant response was from 10 males and 6 females. Three males were excluded due to being outside the age of the inclusion criteria. At Time 2 only 10 participants responded with 3 males lost to attrition and one male excluded due to an incomplete dataset. Thus, the final sample for analysis comprised of 7 females and 2 males.

Page 7 Line 163

The word significant needs to be replaced in this sentence as the design does not allow for statements such as this to be made.

 Thank you for highlighting this claim in our results section. We acknowledge and have removed the word significant on page 7.

Line 173

Other measures that in line 173 they state as being indicators of successful and unsuccessful transitions.

 This sentence has been tempered to include the word ‘potential’ on page 7.

Can two open ended questions added to the About Me survey really assess changes in quality of life

 The Quality of Life Questionnaire was used to measure QoL as per the questions in the different domains. The addition of two open ended questions allowed participants to provide additional information regarding their experiences over the past twelve months. This also allowed participants who wished to explain their QoL experiences in more depth. These were not used to measure QoL, rather to inform case descriptions and identification of factors related to the positive experiences and challenges faced during this transition period. We have amended the description in text on page 9.

Provide support for cut scores of 79 and 80 There are no published clinical cut-offs for the QoLQ, however, according to the Standardisation Manual (Schalock & Keith, 2004) individuals with disability in semi-independent or independent living, engaged in employment, with increased community integration, with a total QoL score of 80 and above are identified as having a high QoL. Those individuals with disability in supervised accommodation, unemployed, with low levels of satisfaction, and a total QoL score of 79 and below are identified as having a low QoL (Schalock & Keith, 2004). 

In order to examine differences during the transition period for participants who showed deterioration and improvement in levels of QoL, in consultation with the instrument authors, a median split of the sample based on total QoL scores was conducted at follow-up (K. Keith, R. Schalock, personal communication, 16 June 2017). Thus, based on the Standardisation Manual (Schalock & Keith, 2004), high QoL was defined as a total QoL score of 80 and above, and low QoL was defined as a total QoL score of 79 and below. We have provided this additional information on page 10 of the text under the Quality of Life questionnaire.

Reanalyse the results using subscale scores of the Quality of Life questionnaire We agree that an analysis of the subscales would provide a comprehensive presentation the transition experience and quality of life outcomes in the different domains. As such, we have completed a reanalysis and included our findings on pages 19, 20 and 21.

---

## [Decision Letter · Decision Letter 1]

20 Apr 2022

Transition approaches for autistic young adults: a case series study

PONE-D-21-38878R1

Dear Dr. Pillay,

We’re pleased to inform you that your manuscript has been judged scientifically suitable for publication and will be formally accepted for publication once it meets all outstanding technical requirements.

Kind regards,

Amanda A. Webster

Academic Editor

PLOS ONE

Additional Editor Comments (optional):

Reviewers' comments:

Reviewer's Responses to Questions

**Comments to the Author**

1. If the authors have adequately addressed your comments raised in a previous round of review and you feel that this manuscript is now acceptable for publication, you may indicate that here to bypass the “Comments to the Author” section, enter your conflict of interest statement in the “Confidential to Editor” section, and submit your "Accept" recommendation.

Reviewer #1: All comments have been addressed

Reviewer #2: All comments have been addressed

2. Is the manuscript technically sound, and do the data support the conclusions?

Reviewer #1: Yes

Reviewer #2: Yes

3. Has the statistical analysis been performed appropriately and rigorously? 

Reviewer #1: Yes

Reviewer #2: Yes

4. Have the authors made all data underlying the findings in their manuscript fully available?

Reviewer #1: Yes

Reviewer #2: Yes

5. Is the manuscript presented in an intelligible fashion and written in standard English?

Reviewer #1: Yes

Reviewer #2: Yes

6. Review Comments to the Author

Reviewer #1: Thank you for your careful consideration and response to comments.

This is a very interesting and important paper that highlights an area which is under researched but which can make a significant difference to the lives of autistic individuals.

Reviewer #2: (No Response)

7. PLOS authors have the option to publish the peer review history of their article (what does this mean?). If published, this will include your full peer review and any attached files.

Reviewer #1: No

Reviewer #2: No

---

## [Editor Report · Acceptance letter]

25 Apr 2022

PONE-D-21-38878R1 

Transition approaches for autistic young adults: a case series study 

Dear Dr. Pillay:

I'm pleased to inform you that your manuscript has been deemed suitable for publication in PLOS ONE. Congratulations! Your manuscript is now with our production department. 

Kind regards, 

on behalf of

Dr. Amanda A. Webster 

Academic Editor

PLOS ONE